# Responses to COVID-19 in Higher Education: Social Media Usage for Sustaining Formal Academic Communication in Developing Countries

**Abu Elnasr E. Sobaih** [1,2,*] **, Ahmed M. Hasanein** [2] **and Ahmed E. Abu Elnasr** [3]

1   College of Business Administration, King Faisal University, P.O. Box 400, Al-Ahsa, Saudi Arabia
2   Faculty of Tourism and Hotel Management, Helwan University, P.O. Box 12612, Cairo, Egypt; ahmed_hasanein@fth.helwan.edu.eg
3   Higher Institute for Specific Studies, Future Academy, P.O. Box 11771, Cairo, Egypt; dr.ahmed.abulnasr@fa-hiss.edu.eg
*   Correspondence: asobaih@kfu.edu.sa

**Abstract:** The worldwide pandemic of COVID-19 has forced higher education institutions to shift from face-to-face to online education. However, many public institutions, especially in developing countries, often do not have access to formal online learning management systems (LMS) for facilitating communication with students and/or among faculty members. This research empirically examines the extent to which social media sites are adopted by faculty members and students for sustaining formal, i.e., sole and official tools, academic communication. For this purpose, online questionnaire surveys, supplemented with in-depth interviews, were undertaken with both faculty members and students. The results showed that students' personal usage of social media has promoted its effective usage for sustaining formal teaching and learning. However, significant differences were found between faculty members and students regarding social media usage for student support and building an online community. Students used social media for building an online community and supporting each other, whereas faculty members were focused on teaching and learning exclusively. The results confirm that proper usage of social media could promote a new era of social learning, social presence and an alternative platform to foster online learning. Research implications for higher education policymakers, especially in developing countries, and scholars are discussed.

**Keywords:** COVID-19; formal academic communication; online learning; online teaching; social media; social presence; higher education

## 1. Introduction

The worldwide pandemic of COVID-19 has presented unparalleled challenges to traditional or face-to-face education. With the need to contain the virus outbreak, countries have implemented measures to reduce gatherings of large crowds and ensure physical social distancing. Thus, most governments applied the quarantine times that led to stopping traditional education. In consequence, governments have shifted all classes from face-to-face to online. However, public universities in many underdeveloped nations, e.g., Egypt, are suffering from a lack of technological platforms and formal online learning management systems (LMS) for communication with students or with their faculty members. Hence, they do not have the full capabilities to support the online learning process [1,2].

The pandemic of COVID-19 has pushed policy makers, university leaders and institution deans in higher education to search for alternatives to the traditionally-based learning system of the physical classroom. Various universities in Egypt, for instance, have encouraged their faculty members to use free communication platforms, e.g., Google Classroom and Zoom. Social media, e.g., Facebook,

WhatsApp and YouTube are formally used by institutions and their faculty members to communicate with their students. Institutions have encouraged their faculty members to communicate with their students via official pages and formal groups on these social networking sites (SNSs), such as Facebook and WhatsApp.

As a consequence of COVID-19, and for the first time, both faculty members and students in many developing countries are forced to officially communicate online for academic-related purposes. With the absence of online LMS, social media can create a great opportunity for these institutions to officially communicate with their students to foster online learning [3,4]. The research builds on the presence of students and academic staff on these social sites to foster online social interaction and create effective online learning experience [3,5].

Previous studies on social media usage in higher education [3–6] have shown that they can be used for supporting communication with students, e.g., to supplement traditional learning and enhance their learning experience. This research, however, makes an attempt to investigate social media usage for sustaining formal academic communication, especially after the COVID-19 worldwide pandemic, in public universities that do not have a robust use of LMS and were reliant on in-class communication prior to COVID-19. In this research, formal academic communication means using social media as a sole and official platform for academic-related purposes, which include teaching and learning, student support, building an online community and program marketing and promotion, after their approval from institution leadership. More specifically, the research examines the extent to which social media are adopted by faculty members and students for sustaining formal academic communication. The guiding research questions for this research were:

1.  What is the extent to which faculty members and students in public higher education institutions use social media for sustaining formal academic communication after the COVID-19 worldwide pandemic?
2.  Are there any differences between faculty members and students in social media usage as a formal communication tool?
3.  How does the use of social media impact on faculty members' and students' patterns and practices of formal academic communication?
4.  What challenges are faced by both faculty members and students in social media usage for sustaining formal academic communication?
5.  How could these challenges be overcome for better social media usage for sustaining formal academic communication?
6.  What lessons and implications could be learned for supporting social presence and fostering the sustainability of online learning using social media?

## 2. Review of Literature

### 2.1. Social Media Usage in Higher Education

Social media have emerged as powerful platforms for possibly enhancing students' learning, facilitating interactions between students and their instructors as well as with their peers, and engaging them in the new distanced learning environment [3,4,7]. Research has also shown that faculty members are using social media for professional and teaching purposes [5]. The top-ranked social media for academic communication are Facebook, WhatsApp, YouTube and Wikipedia [8]. A recent study found that Facebook and WhatsApp are the most used tools in higher education for different academic-related purposes [4].

Several studies [4,9–11] have focused on the broad advantages of social media usage in higher education. Such studies have confirmed the value of social media tools for informal scholarly communication, connectivity, community building, maintaining trust and satisfaction as well as developing students' social-life. Studies also showed the value of social media usage for student engagement and influencing positive student learning experiences [5,12].

### 2.2. Student Perceptions of Social Media Usage for Academic Communication

Several studies [5,13–16] have been conducted to measure the effectiveness of social media tools for improving student integration in higher education. Studies showed that social media, such as Facebook, is considered an effective tool for improving students' performance [13,16,17], increasing students' engagement [5] and improving student awareness of their learning experience [15,18]. Furthermore, there is a direct relationship between the students' educational performance and the usage rate of Facebook for learning [17]. However, other studies [14,19] showed that Facebook has been associated with students' negative educational performance. Excessive use of Facebook was a negatively noteworthy predictor for student engagement [14]. Another study [5] showed that the use of Facebook in learning has created positive student learning experiences.

A study on the efficiency of SNSs to improve the learning experience showed that students found social media as inspiring their learning and promoting active teamwork with colleagues and academic staff [20]. Moreover, amongst the key elements of a student's readiness to use social media for learning, shared resources were very effective. The study also revealed that collaboration was the farthest vital interpreter for usage [21].

### 2.3. Faculty member Perceptions of Social Media Usage for Academic Communication

Studies [5,11,14,22,23] found that using SNSs positively influences learning outcomes as they allow faculty members to engage their students, create knowledge, share and collaborate with each other. Although faculty members have acknowledged the prospective significance of SNSs for academic communication, its practical usage was minimal due to several barriers or challenges [1]. These barriers vary and include digital divide, intimacy and security, loss of control and monitoring, limited support from institutions, limited awareness about the role of SNS as a learning platform and assumptions made by faculty members about appropriateness of SNSs for learning purposes as well as poor IT support and infrastructure in some institutions [1,22].

## 3. Research Methodology

### 3.1. Population and Sample

This study is concerned with public higher education institutions in underdeveloped nations. For pragmatic reasons, the research is concerned with the nine public colleges that grant tourism and hotel bachelor degrees in Egypt. These institutions depended solely on in-class learning and face-to-face communication with students or among faculty members prior to COVID-19 [1,3]. They currently have 597 full-time faculty members at different ranks and about 12,000 students in undergraduate programs. Faculty members include teaching assistants (207), professors (310) and administrators who hold administrative positions (80). For the first phase of the research, which includes a questionnaire survey directed to faculty members and their students (as discussed in Section 3.2), the same framework of sampling in similar research [1] was adopted. The collected and valid questionnaires for analysis were 309 forms from students and 304 forms from faculty members. With regard to interview participants, the number of student and faculty member participants was decided after data saturation was achieved [24]. Hence, 24 faculty members and 30 students were found to be enough to achieve data saturation. There was an equal participation from males and females in the interviews.

### 3.2. Data collection Methods and Procedures

To answer the research questions and achieve aim of the study, a sequential explanatory mixed methodology was adopted with two distinct phases of research [25]. The research started with an online survey directed to both faculty members and students at the nine institutions. The results of the survey informed the second phase of the study. Hence, the survey was supplemented with in-depth, online, one-to-one interviews with faculty members and students.

For the first phase of the study, a pretested questionnaire survey was used to find an answer to the first three questions (i.e., questions: one, two and three). For the second phase of the study, in-depth, online interviews were conducted with a number of faculty members and students using the most convenient online tool for them. Some interviews preferred Zoom or Google Duo, others favored WhatsApp or Facebook Messenger. The purpose of interviews was to find an answer to three research questions (i.e., questions: three, four and five). Two major themes were discussed with interviewees: the actual social media usage for sustaining formal academic communication, especially after COVID-19 pandemic, and challenges facing them during social media use for sustaining formal academic communication. All interviews were one-to-one and each interview was about an hour in length. Interviews were conducted first with 30 students and followed by interviews with 24 faculty members. As highlighted earlier, this number was enough to achieve data saturation. All interviews were voice-recorded after interviewees' verbal and/or written consent. They were all informed before the beginning of interviews that the interview was for research purposes. All interviews were conducted in Arabic language. The scripts were reviewed by two bilingual experts.

*3.3. The Research Instrument*

The questionnaire consisted of four sections. Section one included demographic data, i.e., gender, age, current position for faculty members and gender, age, study level for students. Section two consisted of three points: (a) if they use social media for formal academic interaction; (b) frequency of their usage; (c) type of social media used. Section three assessed actual social media usage for formal academic communication: teaching and learning (8 factors); student support (6 factors); building an online community and connection (4 factors) and program marketing and promotion (4 factors). These factors were adopted from previous research [1]. Respondents were asked to assess social media usage by a 5-point Likert scale (where 1 = no usage, 2 = minimal usage, 3 = moderate usage, 4 = substantial usage, 5 = great usage). Section four asked the participants to add any further comments related to the research.

The questionnaire was in English and Arabic languages. The English version was translated, and two experts in bi-languages checked and approved the translation. In addition, the questionnaire was piloted for 20 faculty members and 30 students to ensure appropriate wording, and areas were checked for improvements. As a result of piloting, a few, but not major, changes were made in wording to fit with students and faculty members. For example, a new heading was added "*social media are used to . . .* ". As another example, some items were reworded, e.g., "*students can check class assignment and receive course announcements*", became "*post class assignment and send course announcements*" in the faculty members' questionnaire and "*check class assignment and receive course announcements*" in the students' questionnaire.

The methodology for the online survey proposed in the literature [26] was followed. Once the instrument was developed, one member of the research team started designing the online survey, and it was properly checked by other team members for presentation and accuracy before sending the URL to participants. An introduction was written to clarify the purpose of the research and invite faculty members or students to participate in this research. Participants were informed of confidentiality and told that the study was for research purposes. The introduction with the URL (English and Arabic) was sent to both faculty members and students in the nine institutions by personal emails and/or via different social media accounts. The research team checked and followed the responses several times daily. There were contact details (i.e., name, telephone, email and social media accounts) added by the end of the introduction for any further enquiries. The reliability of the measures was ensured using Cronbach's alpha which was above 0.70 for all items [27], i.e., 0.91 for teaching and learning; 0.89 for student support; 0.90 for building an online community and connection; 0.89 for program marketing and promotion.

### 3.4. Data Analysis

For the survey analysis, frequencies and percentages were adopted to analyze the respondents' profile. Descriptive statistics, i.e., mean and standard deviation, were adopted to analyze the scale items. An Independent Samples t-test was adopted to compare actual usage of faculty members and students. To examine the magnitude of the differences between faculty members and their students regarding actual social media usage for formal communication, eta squared was adopted. The questionnaire was analyzed using SPSS version 25. All qualitative data were analyzed manually by qualitative content analysis [28]. All recorded interviews were transcribed straight after the interviews, and the language was checked by two bilingual (Arabic/English) experts.

## 4. Results of Survey

### 4.1. The Profile of Faculty Members

There was an equal participation from male and female faculty members in this study, with only a higher proportion of males (61%) holding managerial positions (Table 1). As expected, most of the young academic staff were under 30 years of age (84%). Moreover, professors were between 30 and 40 years old (66.5%), and faculty members who held managerial jobs were in the range of 41 and 60 years of age (83.3%) (Table 1).

**Table 1.** The profile of faculty members *.

| | | Teaching Assistants | Professors | Administrators |
|---|---|---|---|---|
| Gender | Male | 59 (47.2%) | 82 (50.9%) | 11 (61.1%) |
| | Female | 66 (52.8%) | 79 (49.1%) | 7 (38.9%) |
| Age | Less than 30 years | 105 (84%) | 4 (2.5%) | - |
| | 30 to 40 years | 20 (16%) | 107 (66.5%) | 3 (16.7%) |
| | 41 to 60 years | - | 50 (31%) | 15 (83.3%) |
| | Over 60 years | - | - | - |
| The use of social media for formal academic communication | Yes | 125 (100%) | 161 (100%) | 18 (100%) |
| | No | 0 (0%) | 0 (0%) | 0 (0%) |
| Frequency of using social media for formal academic communication | Daily | 125 (100%) | 161 (100%) | 18 (100%) |
| | Weekly | 0 (0%) | 0 (0%) | 0 (0%) |
| Social media used for formal academic communication | Facebook | 125 (100%) | 155 (96.3%) | 16 (88.9%) |
| | WhatsApp | 120 (96%) | 130 (80.7%) | 18 (100%) |
| | LinkedIn | 10 (8%) | 90 (55.9%) | 15 (83.3%) |
| | YouTube | 40 (32%) | 25 (15.5%) | 2 (11.1%) |
| | Others (e.g., Academia) | 15 (12%) | 10 (7.9%) | 2 (11.1%) |

* N = 304.

All faculty members in the study, interestingly, adopted social media for formal academic communication on a daily basis. Facebook was the most frequently used social media for formal academic communication by teaching assistants (100%) and professors (96%), whereas only 88% of administrators used Facebook for formal academic communication, especially with students. As Table 1 shows, all administrators used WhatsApp for formal academic communication with students and other colleagues. WhatsApp was also highly used by other academic staff (Table 1). Other types of SNSs adopted by faculty members for communication with other colleagues or other academics include LinkedIn, YouTube and Academia (Table 1).

### 4.2. The Profile of Students

The ratio of male student participants in this research was almost the same as female participants (Table 2). Interestingly, all participants in this study were relatively young, i.e., under 30 years of age, since there is no a gap between secondary school and higher education in Egypt, and hence, almost all bachelor students are often less than 25 years old [3]. In addition, there was also an almost equal participation from students at different study levels (Table 2).

**Table 2.** The Profile of Student Respondents *.

|  |  | Frequency | Percentage |
|---|---|---|---|
| Gender | Male | 156 | 50.5 |
|  | Female | 153 | 49.5 |
| Age | Less than 20 years | 140 | 45.3 |
|  | 20 to 25 years | 159 | 51.5 |
|  | 25 to 30 years | 10 | 3.2 |
| Study level | Freshman (year one) | 65 | 21.1 |
|  | Sophomore (year two) | 77 | 24.9 |
|  | Junior (year three) | 89 | 28.8 |
|  | Senior (year four) | 78 | 25.2 |
| The use of social media for formal academic communication | Yes | 309 | 100 |
|  | No | 0 | 0 |
| Frequency of using social media for formal academic communication | Daily | 309 | 100 |
|  | Weekly | 0 | 0 |
| Social media used for formal academic communication | Facebook | 309 | 100 |
|  | WhatsApp | 298 | 69.4 |
|  | LinkedIn | 20 | 6.5 |
|  | YouTube | 195 | 63.1 |
|  | Wikis | 155 | 50.2 |
|  | Others (e.g., Blogs and Twitter) | 60 | 19.4 |

* N = 309.

It is interesting that all students who participated in this study used social media for formal academic communication on a daily basis (Table 2). Facebook was the most adopted tool for formal academic communication by students. This was followed by WhatsApp (69.4%), YouTube (63.1%), Wikis (50.2) and other social media (19.4%), e.g., blogs and Twitter (Table 2). A low proportion of students used LinkedIn for academic communication (6.5%).

### 4.3. Social Media Usage for Sustaining Formal Academic Communication

As Table 3 shows both faculty members and students reported almost the same high mean score for social media usage as a learning platform, this means that they both agreed informal usage of SNSs, especially Facebook and WhatsApp, as a learning platform. The results also showed that both faculty members and students had nearly the same low mean score for program marketing and promotion, which means that they both agreed that they were less likely to adopt social media for program marketing and promotion. The low mean regarding social media use for program marketing and promotion at this stage seems typical, as the priority of senior management and students is to ensure the continuance of current term study after COVID-19 pandemic. However, students reported higher mean scores and lower standard deviation in relation to student support, building community and connections than their faculty members. This means that student usage of social media for student support and building connection is higher than their faculty members, whereas faculty members often focus on social media usage to support the formal learning process.

**Table 3.** The Results of Descriptive Statistics.

| Social Media Sites Are Used to … | Faculty Members * | | Students ** | |
|---|---|---|---|---|
| **Teaching and learning** | μ | σ | μ | σ |
| 1. Communicate with and engage students in the courses | 4.421 | 0.6944 | 4.418 | 0.6911 |
| 2. Post/check class assignment and receive/send course announcements | 4.464 | 0.6014 | 4.466 | 0.6000 |
| 3. Create stronger learning communities | 4.234 | 0.7630 | 4.233 | 0.7586 |
| 4. Post/check online lectures (live or recorded) | 4.480 | 0.6342 | 4.459 | 0.6313 |
| 5. Post/check useful academic videos, links and supporting materials | 4.378 | 0.5788 | 4.385 | 0.5788 |
| 6. Facilitate online discussions related to assignments and/or projects | 4.237 | 0.6370 | 4.269 | 0.6464 |
| 7. Post/check students' academic accomplishment or achievements | 4.303 | 0.6192 | 4.311 | 0.6194 |
| 8. Post/answer comments and enquires on academic issues | 4.418 | 0.6078 | 4.388 | 0.6014 |
| **Student support** | μ | σ | μ | σ |
| 9. Provide support and motivate students | 3.158 | 0.7759 | 4.366 | 0.5456 |
| 10. Provide emotional support for students | 3.211 | 0.7502 | 4.314 | 0.6308 |
| 11. Resolve issues related to students | 3.194 | 0.7783 | 4.372 | 0.5872 |
| 12. Provide mentoring for students | 3.250 | 0.7772 | 4.414 | 0.5611 |
| 13. Help to integrate students in work groups | 3.207 | 0.7834 | 4.366 | 0.5913 |
| **Building community and connection** | μ | σ | μ | σ |
| 14. Build and strengthen online community | 2.382 | 0.7659 | 4.252 | 0.5415 |
| 15. Facilitate students' involvement and participation in activities | 2.632 | 0.8497 | 4.233 | 0.5733 |
| 16. Encourage students to share their social activities online | 2.418 | 0.6698 | 4.447 | 0.5932 |
| 17. Connect students with alumni (graduates) | 2.493 | 0.7490 | 4.388 | 0.5388 |
| **Program marketing and promotion** | μ | σ | μ | σ |
| 18. Promote college courses and programs | 2.444 | 0.8657 | 2.434 | 0.8639 |
| 19. Recruit students into specific academic program/courses | 2.507 | 0.8124 | 2.505 | 0.8552 |
| 20. Advertise new programs and courses | 2.477 | 0.6445 | 2.479 | 0.6769 |
| 21. Post/check the seminars and workshops | 2.405 | 0.6971 | 2.385 | 0.6913 |

\* N = 304; ** N = 309.

An Independent Samples t-test was conducted to compare faculty members and students in their usage of social media for formal communication (Table 4). The results showed statistically significant differences between faculty members and students in relation to student support ($t = -46.42$, $p < 0.001$) and building community and connection ($t = -66.98$, $p < 0.001$). However, no significant differences were found between faculty members and students regarding teaching and learning ($t = 0.037$, $p > 0.05$) and program marketing and promotion ($t = 0.231$, $p > 0.05$). The results of eta-squared statistics were more than 0.08 for student support ($\eta2 = 3.756$) and building community and connection ($\eta2 = 5.417$) which indicates a very large effect size between students and their faculty members (Table 4).

**Table 4.** The Results of Independent Samples t-test and Eta Squared.

| | Group | μ | σ | t | Df | p * | η2 |
|---|---|---|---|---|---|---|---|
| **1. Teaching and learning** | Faculty members * | 34.934 | 1.8191 | 0.037 | 611 | 0.970 | - |
| | Students ** | 34.929 | 1.7932 | | | | |
| **2. Student support** | Faculty members | 16.019 | 1.8038 | −46.42 | 536 | 0.000 | 3.756 |
| | Students | 21.832 | 1.2396 | | | | |
| **3. Building community and connection** | Faculty members | 9.9243 | 1.5343 | −66.98 | 567 | 0.000 | 5.417 |
| | Students | 17.320 | 1.1723 | | | | |
| **4. Program marketing and promotion** | Faculty members | 9.8322 | 1.5925 | 0.231 | 611 | 0.818 | - |
| | Students | 9.8026 | 1.5883 | | | | |

\* N = 304; ** N = 309; η2 = eta squared (Cohen's d) = (M2 − M1)/SD pooled; d ≤ 0.2 small differences; d = 0.5–0.8 medium differences; d ≥ 0.8 large differences.

## 5. Results of Interviews

The results of the interviews with faculty members showed that they received a call from their institution leadership to connect with their students online as a response to COVID-19 and the stopping of traditional face-to-face learning. Interestingly, all faculty members stated that they were flexible to choose the most convenient free tool for online communication with their students. All interviewed faculty members also agreed that the call did not include a specific type of online tool or social media. With limited support from institutions, faculty members started their self-learning about online platform usage. Surprisingly, faculty members found that they could not reach their students without social media accounts due to the absence of formal LMS. Hence, they developed a formal closed group for each course to connect with their students. Institutions advised faculty members to contact students via the student group leader (who is often one of the active students, e.g., in student groups on campus or on the student council) (Figure 1). The results showed that some faculty members (9 out of 24) argued that their institutions created the social media group and cascaded the group link between faculty members and students via group leaders (Figure 1).

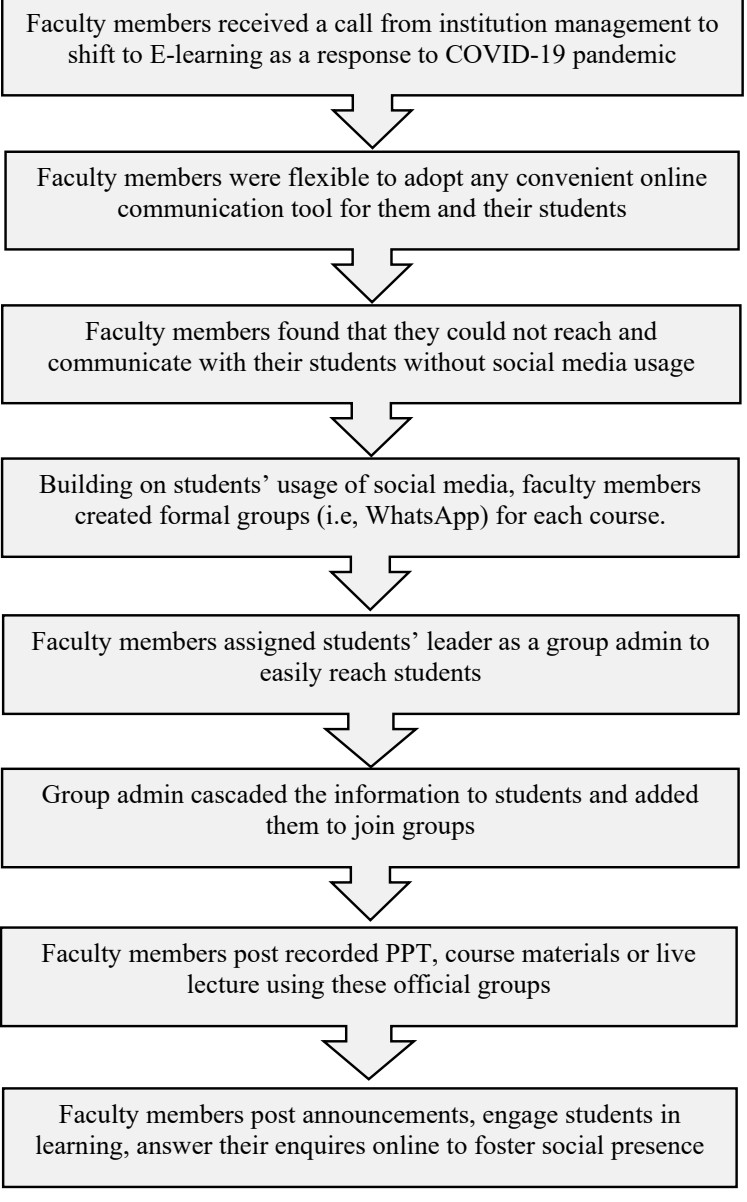

**Figure 1.** Social media usage by faculty members for formal academic communication with students.

Despite few faculty members (5 out of 24) stating that they used online meeting platforms, e.g., Google Classroom and Zoom. Several barriers were identified by faculty members regarding these platforms, e.g., many students were unfamiliar with these platforms since they did not receive any training. As a result, faculty members said they also had to switch to social media groups for formal communications with their students—mainly Facebook or WhatsApp. The results showed that most faculty members (19 out of 24) used social media groups as the sole tool for academic communication with their students, while others faculty members (5 out of 24) used these social media groups as well as free online meeting platforms, especially Google Classroom and Zoom.

Faculty members agreed that there were several ways to adopt these groups of SNSs for formal learning purposes. Many faculty members (17 out of 24) said they recorded PPT or uploaded the course material as a PDF document supplemented with voice notes (Figure 1). Other faculty members offered live lecturing on a weekly basis at the same scheduled time of traditional lectures. Few faculty members (2 out of 24) developed a YouTube channel and posted video lectures for their students.

With this new experience of online learning, using social media, all interviewed faculty members agreed it is a great experience for them, and social media helped them in achieving their course intended learning outcomes. However, a considerable number of interviewed faculty members (10 out of 24) argued that social media added more stress for them with the overwhelming questions and enquiries from students throughout the day. There was also consensus among interviewed faculty members that the primary goal of these social media groups is providing a proper learning experience of the same value as traditional face-to-face learning. Therefore, faculty members argued that they spend all their efforts to answer students' questions and enquiries as well as engage them as much as possible in this social learning environment.

Interestingly, all interviewed faculty members agreed that they used WhatsApp for informal communication before the COVID-19 pandemic. However, after the pandemic, WhatsApp groups became the formal communication between them and other staff at their institutions. Faculty members also revealed that WhatsApp Web effectively facilitates document sharing, gaining feedback, and holding a meeting.

Supporting the results of the survey, especially the gap between faculty members and students, the results of interviews with students showed that faculty members focus solely on supporting formal learning, and limited, if any, attention is paid to student support and building a proper online community. However, the vast majority of interviewed students (25 out of 30) agreed that social media properly engaged them in learning via communication with their counterparts in asocial learning environment. Almost all students (28 out of 30) highly appreciated social media usage for formal teaching and learning compared to other online tools, e.g., Google Classroom due to its perceived ease of use, interactivity and usefulness. There was consensus among students, especially freshmen and sophomore, that if faculty members pay better attention to student support and building an online community, social media will be a more appropriate tool for formal academic communication compared to face-to-face classroom learning. Some freshmen argued that they are interested in more emotional support from their colleagues as well as their faculty members to ensure their engagement in learning due to this new online experience. Faculty members and their students raised numerous challenges that affect social media usage for formal academic communication, as will be discussed in the next section.

*Challenges and Recommendations of Social Media Usage for Sustaining Formal Academic Communication*

Both faculty members and students raised a number of challenges that should be overcome for proper adoption of social media for formal academic communication. The results of interviews with faculty members and students showed 15 challenges during their usage of social media for sustaining formal academic communication (Table 3). Three challenges were related to institution, i.e., learning policy and plan, management support and IT infrastructure. There was an agreement between both faculty members and students that the call for shifting to online learning was not supported by a clear

policy and action plan. Hence, the route was not fully clear for both faculty members and students. There were no published full guidelines or a toolkit provided to faculty members and students on how this shift in learning could be properly undertaken. This was supplemented by limited management support, poor IT infrastructure and absence of professional IT support, which added to the sufferance of faculty members and students. There were two challenges connected to students, i.e., student support and shared course material and notes. Students agreed that the priority of faculty members for social media usage is supporting formal learning, but limited attention is given to student support and building an online community. This was crucial for students, especially those at the first levels of study, i.e., freshmen and sophomores. Some students were also concerned about the quality of materials and voice notes. They argued that presentations of some material and voice notes sometimes were not clear enough. On the other side, faculty members were concerned about another two challenges, i.e., types of used social media and delivering practical courses. Faculty members argued that institutions were flexible about the used tool for communication with students; however, this added more stress for them since students started communication with them using different tools, e.g., Facebook and WhatsApp, at the same time. Faculty members also found a great difficulty in teaching practical courses using social media, which often requires physical evidence and attendance. The majority of faculty members (16 out of 24) stated that practical courses such as cooking, foodservice, housekeeping or practical museum cannot be taught online or with social media tools.

The results also showed four overlapped challenges between institutions, faculty members and students. The first challenge was related to the online learning culture, which was new for institutions, faculty members and students in public institutions. The COVID-19 pandemic did not give a chance for them to prepare themselves properly, which was the case at all other public institutions. With this new culture, the absence of ethical codes and class observation from institutions, there was an inappropriate street language using slang and practices that affect the quality of communication between faculty members and students. There was an agreement among faculty members that social media cannot be used for assessing and grading students, i.e., quizzes and exams. Regarding the awareness of social media usage for academic communication, some students (9 out 30) and faculty members (7 out 24) were not fully aware of social media services which can help in formal academic communication, e.g., WhatsApp Web and live videos on Facebook.

There were two challenges interlinked between faculty members and their students. The first challenge was that students can post enquiries and ask questions at any time of the day. However, faculty members did not have the time to respond to each question on time because they can exceed hundreds daily in each course. There was also a concern from both faculty members and students about their privacy and account security since they have access to their personal phone number or friends on Facebook. Some faculty members (5 out of 24) were very concerned that their students can share their personal lives. Other challenges raised by faculty members were related to the coordination with students; many institutions have chosen a group leader from the students to coordinate between faculty members and students. Most faculty members (17 out of 24) did not like this practice and argued that once a group has been created and students were added, the group admin should be either a faculty member directly or one of the teaching assistants. The last challenge was related to variation in students' needs and expectations. Students have different learning styles and preferences. While some students prefer documents with voice notes, others prefer live lecturing or video showing. These varieties in needs should be addressed by institutions and their faculty members. Recommendations were suggested and discussed with interviewees to overcome all challenges raised by them (Table 5).

**Table 5.** Challenges and recommendations for social media usage as a formal academic tool in higher education.

| Challenges | Examples | Recommendations |
|---|---|---|
| **Learning policy and plan** | - Unclear policy for social media usage as a formal academic tool<br>- No announced plan on how to use social media for formal academic communication | - Institution leaders should establish and announce a clear policy for social media usage as a formal academic tool.<br>- Establish a plan on how to use social media for formal academic communication.<br>- Step-by-step guide and/or toolkit for proper use should be provided by institutions |
| **New learning culture** | - First time for faculty members and students to use social media as a formal academic tool<br>- Inadequate home working environment | - Guidance and support should be provided by the leadership of the institutions for both faculty members and students<br>- Institution leader and family support is needed to ensure proper environment for communication |
| **Management support** | - Lack of class observation<br>- Priority of senior management is teaching and learning | - Online class observations should be undertaken with regular reports on activities to sort out any challenges for faculty members and students<br>- Senior management should ensure student support and building an online community properly |
| **Time constraints** | - Undetermined time for enquires and questions<br>- Time for assignment discussion | - Time to answer enquires and questions as well as interactive discussions should be announced in advance for effective time management by faculty members and students.<br>- Teaching assistants can help in answering students' inquiries and questions out of scheduled times. |
| **Privacy and security** | - Tracking activities<br>- Personal Life | - Faculty members and students can use an official account instead of a personal one, i.e., a new account for formal academic communications on Facebook<br>- Creating a closed group for each course under the administration of teaching assistants |

**Table 5.** *Cont.*

| Challenges | Examples | Recommendations |
|---|---|---|
| **Ethical considerations** | - Inappropriate comments or posts<br>- Slang language | - Establish an ethical code for using the social media for formal academic communications.<br>- Publish the code among both faculty members and students, i.e., all groups of social media.<br>- Assign the control of comments and posts to groups admin, i.e., teaching assistances |
| **IT infrastructure** | - IT support<br>- Poor internet services<br>- Internet cost | - Establish an IT support unit to give guidance/technical support for faculty members and students when needed.<br>- In collaboration with the Ministry of Telecommunication, the higher education policy makers should facilitate internet access for learning purposes |
| **Awareness** | - Lack of awareness by faculty members and/or students on how to use the social media for formal academic communication | - Training session and workshops by institutions on how to properly use social media for formal academic communication.<br>- IT support unit should give guidance/technical support for faculty members and students continuously. |
| **Assessment and grading** | - Social media are inappropriate tools for quizzes and exams | - Use other types and tools for student assessments (e.g., projects and assignments)<br>- In collaboration with Facebook Team, policy makers of higher education can establish new methodologies for student assessment, e.g., exams on social media |
| **Students' needs** | - Variation in students' needs and expectations<br>- Loss of control and monitoring | - Faculty members should provide a variety of teaching methods and tools, e.g., videos, recording, PPT and interactive discussion, to engage students in the learning process.<br>- Faculty members should regularly monitor students' progress with help from teaching assistants. |

**Table 5.** *Cont.*

| Challenges | Examples | Recommendations |
|---|---|---|
| **Coordination with students** | - Inappropriate coordination with students through group leader | - Coordination with students would be performed directly with faculty members or through teaching assistants.<br>- Senior management should monitor coordination and has to be updated with any changes. |
| **Practical courses** | - Social media are not an appropriate tool for practical courses | - Videos with simulating environments could be developed to engage students in the learning of practical courses<br>- Institution management should provide special support to faculty members to develop high quality videos for this purpose |
| **Student Support** | - Lack of interaction with faculty members and delay in response to inquires<br>- Emotional support | - Faculty members should set a predetermined time for discussions and interaction with their students<br>- Fresh and new students should be emotionally supported by faculty members and institutions to ensure their engagement in the online learning community |
| **Shared material and recording** | - Unclear materials and recordings are provided | - Material should be double-checked to ensure its appropriateness for online learning.<br>- Students' comments on materials and recording should be highly considered. |
| **Types of used social media** | - Faculty members are inconsistent in their use of social media tools | - Faculty members, in collocation with institution leaders, should decide the primary social media tool to be used for formal communications instead of using new tool every time. |

## 6. Discussions

There is a growing academic body of literature on social media usage for different purposes in higher education, e.g., supporting the learning process, student support and engagement, scholarly communication and building connections [1,4,6,8,10,19,29]. Notwithstanding, studies were focusing on social media as a supplement instrument for formal online LMS or face-to-face learning. This study is an attempt to examine social media usage for sustaining a formal, i.e., sole and official, academic communication platform in the nine colleges that provide hotel and tourism bachelor degrees after the COVID-19 pandemic and the stopping of traditional classroom learning.

The results showed that Facebook and WhatsApp groups were the most used formal communication tools between faculty members and their students. Moreover, WhatsApp groups were the most used communication tool between faculty members themselves and their institutions. Both faculty members and students effectively used social media for formal online learning. They are at either a substantial or great level of usage. However, significant differences were found between faculty members and students regarding social media usage for student support and building an online community. Students were found to use social media for supporting each other and building an online community and connection, whereas faculty members focus solely on formal learning to meet learning outcomes of their courses. This confirms a gap between students and faculty members in their usage of social media for academic related purposes [1].

Despite the limited interactivity by faculty members on social media groups, i.e., Facebook and WhatsApp, compared to their students, both faculty members and students agreed that this new online communication culture for the first time at the public higher education institutions in Egypt is a great experience for them. Students perceived social media groups as more interactive, easy to use and useful than other free online platforms, i.e., Google Classroom and Zoom, which opens the door for a new era of online learning using SNSs. Students have more patterns and practices on social media than their faculty members who just focus on formal teaching and learning. Students used social media for connection with their faculty members, supporting each other and building an online community.

Supporting the notions made by [30,31] that the use of social media for online learning and academic communication, e.g., Facebook and WhatsApp, could foster social learning and social presence, this research argues that the proper use of social media for formal academic communication could stimulate an interactive learning environment, foster social presence and enhance learning outcomes.

The results of interviews with faculty members and students showed 15 challenges that hinder the appropriate use of social media for formal academic communication. Some of these challenges were related to institutions, whereas others were related to faculty members and students. Furthermore, all these challenges are interlinked (Figure 2). Faculty members and students agreed that the overcoming of these challenges would encourage social media usage for formal academic communication. It would promote a paradigm shift in online learning and shift many programs into online learning using these social media tools. Hence, several recommendations were suggested to defeat these challenges or barriers to appropriate social media usage (Table 5).

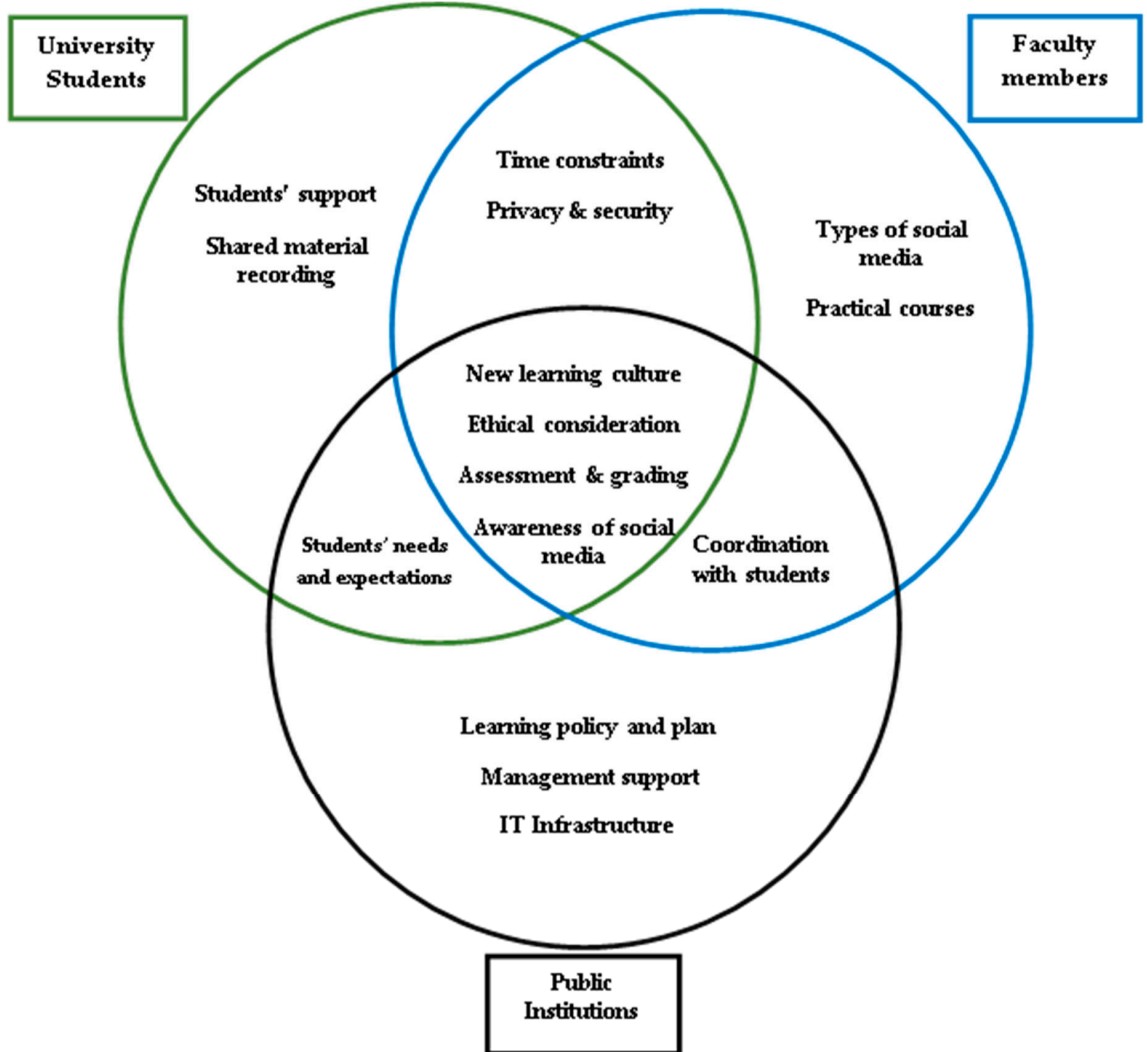

**Figure 2.** The challenges of social media usage for sustaining formal academic communication.

## 7. Implications of the Study

This study has several implications for public higher education policymakers, scholars, faculty members and students, especially in a developing countries context. The policymakers in public higher education in many developing countries have had a quick and good response to the COVID-19 pandemic by adopting a new culture of online learning using free online platforms and social media. However, they have to be proactive and develop an appropriate policy that fosters sustainable online learning. The policy should promote social media usage for formal academic communication. The policy should also be supplemented with an action plan and toolkit on how faculty members and students could use social media for formal communication. It is crucial that policy makers in public higher education, especially in developing countries, collaborate with the telecommunication sector to ensure the provision of appropriate internet services for faculty members and students to support their online learning.

Institution leaders have a vital role in policy implementation and ensuring appropriate support for their faculty members and students. For example, they should establish an IT unit to give guidance and technical support to faculty members and students if needed. Online training and workshops could be provided for faculty members and students by this unit to ensure their appropriate usage of social media and other online platforms. It is also important that they monitor the proper social media

usage via online class observation. They should establish and publish an ethical code among faculty members and students for using social media for formal academic communication.

Faculty members should give more attention to interactivity, student support and building an online community. Special support and attention have to be given for freshmen who are new at university and the online learning experience. Moreover, faculty members also have to check the quality of material and voice notes or narration. Students' feedback has to be acknowledged. Students, on the other hand, have to understand that they should use suitable language duringsocial media formal communication. They should also understand that faculty members have different commitments at their institutions, and the delay in response to their enquires does not mean ignorance or disrespect as they assumed.

The study sends an important message to scholars that more research is needed regarding the potential of social media as a formal academic communication tool. This research sets the first nucleus for social media usage as the sole official tool for academic communication. However, new research could address this paradigm shift in learning using social media as a formal online platform for academic communication. The research could address the influences of social media as a formal academic communication tool on students' learning experience, satisfaction and academic performance. Scholars should recognize the variation in needs and usage of social media for academic communication between faculty members and their students. This research found that social media usage as a formal academic communication tool in higher education supports social learning and social presences and fosters online learning and interaction.

## 8. Conclusions

This research is a response to the impacts of COVID-19 on higher education, particularly public higher education institutions in developing countries, i.e., Egypt. The COVID-19 pandemic has forced institutions to shift from traditional to online learning. Notwithstanding this, many public higher education institutions do not have full access to online LMS [1]. The research is an attempt to investigate the actual use of social media for sustaining formal, i.e., sole and official, academic communication in public higher education in Egypt, using the nine colleges providing tourism and hotel bachelor education as the case study.

The study was set to answer six main research questions. In response to research question number one "what is the extent to which faculty members and students in public higher education institutions in Egypt use social media for sustaining formal academic communication after COVID-19 worldwide pandemic?", the research showed that, for the first time, both faculty members and students adopted SNSs as an official platform for academic communication, especially for formal learning, and the level of usage was at a substantial or great level. Facebook and WhatsApp groups were the most used social media platforms by faculty members and students. Regarding question number two "are there any differences between faculty members and students in social media usage as a formal communication tool?", significant differences were found between faculty members and students in relation to student support and building an online community. Faculty members used social media solely for sustaining formal teaching and learning, whereas students used them for engaging in learning process, supporting each other as well as building an online community. This confirms a variation between students and faculty members in social media usage for formal academic communication. Concerning question number three "how does the use of social media impact on faculty members and students' patterns and practices of formal educational communication?", the research showed that the presence of students and faculty members on social media facilitates communication for formal academic usage. Students perceived social media as a more appropriate tool for communication than other free online communication tools, e.g., ZOOM and Google Classroom due to its ease of use, usefulness and interactivity. Thus, they effectively used social media for a positive learning experience.

With regard to questions four and five "what challenges are faced by both faculty members and students in social media usage for sustaining formal academic communication?; how these challenges

could be overcome for better use of social media for sustaining formal academic communication?", the research showed 15 challenges related to social media usage for formal academic communication (Table 5; Figure 2). Faculty members and students agreed that the overcoming of these challenges would promote proper usage of social media for formal academic communication in higher education, especially with the limitation of free online platforms, e.g., the length of meeting time or number of participants in the meeting.

Regarding question number six "what lessons and implications could be learned for supporting social presence and fostering online learning using social media?, the research supports the notions that the use of social media as formal academic communication supports social learning and social presence [30]. The research confirms that the social relationship between faculty members and students facilitates online communication and interactivity which supports social presence, fosters online social interaction and creates a proper learning environment. It is crucial that stakeholders as well as scholars in higher education, especially in developing countries, recognize this paradigm shift in learning and examine the factors that affect the proper usage of social media for sustaining formal communication in this era of social online learning.

## 9. Limitation and Opportunities for Further Research

This research is concerned with social media usage as a formal tool for academic communication in public higher education institutions in Egypt using institutions offering tourism and hotel degrees as a case study. These institutions do not have a robust use of LMS and werereliant on in-class communication prior to COVID-19. These institutions were extensively dependenton social media for formal academic communication. Hence, the results could be limited due to the tourism discipline, cultures or features of these institutions. Therefore, further research could examine the results in different disciplines within different institutions in the context of various countries.

**Author Contributions:** Conceptualization, A.E.E.S.; data curation, A.E.E.S. and A.M.H. and A.E.A.E.; formal analysis, A.E.E.S. and A.M.H. and A.E.A.E.; investigation, A.M.H. and A.E.A.E.; methodology, A.E.E.S. and A.M.H. and A.E.A.E.; project administration, A.E.E.S.; supervision, A.E.E.S. validation, A.E.E.S. and A.M.H. and A.E.A.E.; writing—original draft, A.M.H. and A.E.A.E.; writing—review and editing, A.E.E.S. All authors have read and agreed to the published version of the manuscript.

**Funding:** The authors acknowledge the Deanship of Scientific Research at King Faisal University for the financial support under Nasher Track (Grant No. 186402).

**Conflicts of Interest:** The authors declare no conflict of interest.

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
