# Peer review of "Responses to COVID-19 in Higher Education: Social Media Usage for Sustaining Formal Academic Communication in Developing Countries"

_sustainability, doi:10.3390/su12166520_

Round 1
Reviewer 1 Report
Dear authors,
I have read your paper with much interest. The topic of the paper titled “Responses to COVID-19 in Higher Education: Social Media Usage for Sustaining Formal Academic Communication in Developing Countries” is relevant. This study presents some rich qualitative data regarding using social media for academic purposes.
The research topic is placed in context and well documented. The gap is highlighted and theoretically substantiated. The novelty of the study is clearly evidenced and referenced to similar studies in the field. The purpose of the research is addressed to research questions. It is clearly mentioned whether the present research complements certain deficiencies existing in recent research in the field. The conclusions answer the aim specified in the study and are supported by results.
However, there are some issues that I noticed could be improved.
Please find below minor points in the article which needs clarification / refinement / reanalysis, rewrites and/or additional information and suggestions for what could be done to improve the article.
Manuscript structure
I suggest to write all sub-titles in italic (see line 113 and 124).
Please pay attention to formatting text (see lines 94-96).
Title
In the title and throughout the manuscript the paper refers to “communication”. Since the research shows that it is not about just communication (it is about more; it is about “teaching and learning. Building community and connection, program marketing and promotion”), if you agree, I recommend to replace "Communication" with "purposes" and to remove "Developing Countries" or replace with "Egypt" in the title. A reformulation suggestion for the title would be: "Responses to COVID-19 in Higher Education: Social Media Usage in Public Universities to Sustain Educational Purposes” or other rephrasing based on above mentions. If you agree this, then the term of "communication" must be clarify and replaced throughout the manuscript.
Other alternative can be to keep "Academic Communication" and to explain in the introduction what represents/includes "Academic communication" in context of this study.
Abstract
I recommend to replace the “faculty” with “faculty staff” or “faculty members” (as it is write in line 117), both in the abstract and throughout the manuscript.
Line 19: If you agree, I recommend to remove “sites”.
Line 25: If you agree, I recommend to add “online teaching”
Introduction
Line 35: I suggest to replace “electronic” with “online”
Line 36: I suggest to remove “e.g. Blackboard”
Line 37: I suggest to keep just “online” and remove “and virtual”
Line 41: I suggest to remove “extensively” OR keep it and add a reference
Line 44: I suggest to remove “especially” OR keep it and add a reference OR replace with “e.g./like”
Line 62: I suggest to numbering the questions
Methodology
Line 118: If you agree, I recommend to replace “professor” with “teacher” OR “university teacher”. If you agree this, then the term of "professor" must be replaced throughout the manuscript (including text and table 1).
Line 120” Please pay attention to the number of students (at line 120, is about 307 students) and correlate with the number from the Table 2 (line 204), where is about 309 students.
Line 130-163: I suggest to put these paragraphs in a separate sub-section, “3.3. Research Instruments”. Then, actually section “3.3. Data Analysis” would be 3.4.
Line 131-135: If you agree, I recommend to write just the questions number (see the comment for the line 62).
Line 164-174: I suggest to detail how the interviews were performed (How many interviews were applied? What was the number of participants for each interview? What was the time for each interview? The interviews were audio/video recorded with permission of the participants?)
Line 167-171: If you agree, I recommend to write just the questions number (see the comment for the line 62).
Line 176-181: I suggest to mention what software was used for (quantitative and qualitative) data analysis
Results
Line 189: I suggest to write the number of the participants in the table legend (N=304)
Line 204: I suggest to write the number of the participants in the table legend (N=309). Pay attention to the comment for the line 120)
Line 224: I suggest to write the number of the participants in the table (“Faculty, N=…, Students, N=..)
Line 234: I suggest to write the number of the participants in the table legend both for students and faculty members and to remove the third column containing this information
Line 248: I suggest to replace “124” with “24”. I think it is about writing error.
Discussion
If possible, it would be interesting to compare the obtained results with more research in the literature that addresses in this subject.
Line 425-426: I suggest do not generalize (“Notwithstanding this, public higher education institutions do not have online LMS.”. If you keep this, I suggest to add a reference.
Author Response
Many thanks for your review and your constructive feedback. Please see a response to all comments in the attached file as well as modified manuscript.

Reviewer 2 Report
- This study used a mixed methodology. It is suggested that section 4 Results could be separated into two sub-parts (using proper headings) in order to be seen easier to follow. Also, the authors can state the selection method of interviewees and/or provide a profile of interviewees if possible.
- Figure 2. “Awareness of social media” is suggested to be classified in students-faculty-institutions interlinked area (middle) as it can be improved with IT supports and training courses provided by institutions. It will involve a collaboration among three parties.
- This study mainly surveyed and interviewed the faculty members, students, and administrators from public colleges that grant tourism and hotel degree in Egypt. The authors may state how the findings of the present study contribute to the community of the chosen field (or even more other fields); or whether the findings are limited due to different subject/discipline/institution cultures or features.
Author Response

(The authors gave the same response as above.)

Reviewer 3 Report
The purpose of the study was to examine how social media is adopted as a medium in times of covid through surveys and interviews.
The results showed that students’ personal usage of social media sites has promoted its effective usage for sustaining formal teaching
and learning. However, significant differences were found between faculty and students regarding social media usage for students’ support and building online community.
A challenge that all universities are facing is that all improvements and technologies has always been developed to facilitate and complement in-person learning, where online learning is seen to be a substitute only when in-person is not feasible. This assumption has driven the difficulty for faculty in adopting online learning in covid, where few alternatives exist from inperson learning. This article is a timely discussion on the issues and challenges faculty will likely encounter in the coming term.
The focus of this study on socia media for sustaining formal academic communications is difficult to discern. While the lit review describes social media use in general, the various types of sm that engages audience in different ways was not defined. Moreover, formal academic communications is also not defined. I suggest these terms should be defined prior to publication. Finally, tools are available such that formal channels in LMS can be channeled and fed to sm if the user chooses, wouldn't such situation meet the need for both formal communication channels and availability in sm?
More information is needed to know about the items. Given the unidimensional nature of the survey of 21 items only yielded a reliability of .7, there may be multiple factors in the responses. Results from the survey also points at an anomalous finding, I'm not certain, it says a majority of faculty members uses Linked in for formal communications. I wonder if this is just a misconception from whether they use linked in because I am skeptical on how it can be done with students.
Results about the uniform use of social media is heartening, but it seems the overwhelming use of facebook may point to either a lack of communication medium or the faculty didn't condone the use of enterprise systems like google or microsoft. The findings that a differnt perspectives on sm use in building community and student support is of note. This may point to the mismatch in resources from an institution point of view, where faculty would imagine those are tasks that require in person interventions when students seek them on an online basis.
I support the publication of this manuscript with the following caveats: 1) I think a limitation of this study is that it applies to institutions that don't have a robust use of LMS and was reliant on in-class communication prior to covid.
2) I think social media is an evolving definition. I think that online classroom and LMS serve to certain extent as a good proxy for communication medium, but in vaccum of these infrastructure, it seems personal sm channels are then used.
3) I think knowing how students and faculty communicated prior to covid would give a better sense of the scale of this adaptation. For example, if students already had their tutorial groups set on whatsapp, then I think the transition during covid seems natural.
But I think this article highlights how whatsapp can be made to work (in their circumstances), but I question the definition of formal academic communication. Surely no one wants their grade to be PM'd.
Author Response

(The authors gave the same response as above.)

Round 2
Reviewer 3 Report
The authors provided sufficient changes to address the concerns I had posed. I now support the publication of this article.